# Relationship between Abdominal Volume Index and Body Adiposity Index and Scales of Insulin Resistance and Metabolic Syndrome

**DOI:** 10.3390/diagnostics13213356

**Published:** 2023-10-31

**Authors:** José Ignacio Ramírez-Manent, Ángel Arturo López-González, Pilar Tomás-Gil, Pere Riutord-Sbert, Lucía Garrido-Sepulveda, María Teofila Vicente-Herrero

**Affiliations:** 1ADEMA-Health Group, IUNICS University of Balearic Islands, 07009 Palma, Spain; joseignacio.ramirez@ibsalut.es (J.I.R.-M.); secretaria@eua.edu.es (P.T.-G.); pereriutord@gmail.com (P.R.-S.); correoteo@gmail.com (M.T.V.-H.); 2Faculty of Medicine, University of the Balearic Islands, 07009 Palma, Spain; 3Institut d’Investigació Sanitària de les Illes Balears (IDISBA), Balearic Islands Health Research Institute Foundation, 07004 Palma, Spain; 4General Practitioner Department, Balearic Islands Health Service, 07003 Palma, Spain; 5Home Hospitalization Unit, Arnau de Vilanova Hospital, 46015 Valencia, Spain; luciags85@gmail.com

**Keywords:** metabolic syndrome, insulin resistance, ROC curve, obesity

## Abstract

Introduction, objectives: Obesity is a global health problem with a great negative impact on health. Among the pathologies caused by obesity are insulin resistance and metabolic syndrome, which constitute an increasingly common health problem in both developed and developing countries. The aim of this study was to examine the relationship between two scales that assess obesity—based on hip circumference—and metabolic syndrome (MetS) and insulin resistance risk scales as predictors of these alterations. Materials, methods: A descriptive, cross-sectional study was carried out on 193,462 workers from different Spanish regions and work groups between January 2019 and September 2021. Abdominal volume index (AVI) and body adiposity index (BAI) were evaluated to assess obesity and its association with insulin resistance using three risk scales (TyG index, Triglycerides/HDL, and METS-IR), while their association with metabolic syndrome was determined using the NCEP ATP III, IDF, and JIS models. Results: The results of the ROC curves to determine the predictive value of BAI and AVI in relation to the three criteria evaluated to calculate MetS in all instances presented a higher area under the curve (AUC) for AVI. The high values of AVI stand out for predicting MetS when applying the IDF criteria. The cut-off point in women was 13.70 with a Youden index of 0.802, whereas in men, the cut-off point was set at 17.59 with a Youden index of 0.672. Regarding the relationship of BAI and AVI with insulin resistance risk scales for both sexes, the AUC only revealed high values when using the METS-IR formula for both AVI and BAI. The AVI cut-off points to predict high values of insulin resistance risk scales in women were established at 13.12 with a Youden index of 0.722. In men, the cut-off point was 17.59, with a Youden index of 0.626. The BAI cut-off points in women were set at 33.88 with a Youden index of 0.748. In men, the cut-off point was 27.91, with a Youden index of 0.598. Conclusions: AVI demonstrated its value as a predictor of metabolic syndrome while exclusively applying the IDF criteria. AVI and BAI demonstrated their value as predictors of high values of insulin resistance risk scales only in the case of METS-IR. This predictive value is also higher in women.

## 1. Introduction

Obesity can be defined as a state in which there is an abnormal or excessive accumulation of fat within the body in relation to the values expected according to height, age, and sex. This accumulation of fat has a negative impact on health, as it is associated with metabolic syndrome and insulin resistance [1,2].

Although there are numerous ways to measure this excess and its distribution in the body, it is essential to use a simple technique that enables the diagnosis and classification of the problem for epidemiological purposes regardless of race and country [3,4], whether to compare populations, to identify individuals at risk of morbidity and mortality [5,6,7], or to establish priorities for individual and community intervention [8,9,10].

Even though populations of various races, ethnicities, and environmental backgrounds are very different in terms of the risk associated with the accumulation of a specific amount of fat or its distribution, it is crucial to establish a consensus on the definition and classification of obesity [11,12]. Determination of this unification is currently based on body mass index (BMI) [13,14] and waist circumference [15,16]. However, there are other scales that evaluate obesity that are based on hip circumference—such as the abdominal volume index (AVI) [17,18]—which better assesses the accumulation of fat in the abdominal area, thereby providing a better prediction for the development of metabolic syndrome (MetS) and cardiovascular risk (CVR). There is also the body adiposity index (BAI), although this does not offer better results than BMI to evaluate metabolic and cardiovascular risk, depending on the origin of the population and race studied, according to published literature [19,20]. This constitutes a challenge for clinical research that must advance on this topic.

Insulin resistance is an increasingly common health problem today [21], and it is rising in both developed and developing countries [22,23]. It appears when the body is not able to respond normally to the insulin produced by the pancreas, which makes it difficult for glucose to enter the cells and be used by them. Its appearance is favored by a high BMI, inflammation, and oxidative stress caused by obesity, diet, age, physical inactivity, and paternal history [24].

Under normal conditions, insulin is involved in the metabolism of carbohydrates, lipids, and proteins and in anabolic function. The decrease or complete loss of the sensitivity of the liver, muscle, or fatty tissue to the action of insulin is what constitutes insulin resistance. [25].

The loss or alteration of the essential action of insulin stimulates an increase in insulin production by the beta cells of the pancreatic islets, which causes an elevated level of insulin in the blood, as well as raised glucose levels. This hyperinsulinemia causes a greater production of fatty acids, which further hinder the action of insulin [26] and cause early atherosclerosis, atherogenic dyslipidemia, and arterial hypertension. This excess of fatty acids accumulates in the abdominal adipose tissue—known as central obesity—in muscles and also in the liver—known as hepatic steatosis [27] or fatty liver.

The fundamental component that facilitates insulin resistance is excess nutrients, which causes ectopic lipid storage that leads to insulin resistance at the muscle and liver level and the consequent increase in adipose tissue [28,29,30,31]. This increase in adipose tissue may not be pathological as it occurs in adolescence, aging, or even pregnancy, but it is closely related to MetS (hypertension, dyslipidemia, obesity, and diabetes), although they are not synonymous [32,33,34].

Based on the above, the objective of this study was to examine the relationship between two scales that assess obesity—based on hip circumference—and MetS and insulin resistance risk scales as predictors of these alterations.

## 2. Methods

A descriptive, cross-sectional study was carried out on 193,462 workers (116,407 men and 77,055 women) from different Spanish regions and work groups, especially hospitality, construction, commerce, health, public administration, transport, education, industry, and cleaning.

The workers were selected from those who attended the medical health examinations carried out in the different participating companies in the period between January 2019 and September 2021.

The following were established as inclusion criteria (Figure 1):-Being between 18 and 67 years old.-Belonging to one of the companies participating in the study.-Not drinking alcohol regularly.-Agreeing to participate in the study and granting the use of the data for epidemiological purposes.

Health professionals from each participating company determined the anthropometric, analytical, and clinical variables necessary to calculate the scales used in this study. Measurement techniques were standardized to reduce potential biases during the process of obtaining variables.

A SECA model height and weight scale was used to measure the height and weight of the person while they were in an erect position with their abdomen relaxed, according to the international standards for ISAK anthropometric evaluation [35]. Abdominal waist circumference was measured in this same position, with a measuring tape placed parallel to the floor at the midpoint between the last rib and the iliac crest [36]. To measure hip circumference, the subject had to be standing and have their arms crossed over their chest. The measuring tape was placed on the most prominent part of the buttocks, making sure that it was completely horizontal (at the same height) at the back and front.

Body mass index (BMI) was obtained by dividing weight (in kg) by height (in meters) squared.

A calibrated OMRON M3 automatic sphygmomanometer was used to measure blood pressure in the supine position after 10 min of rest (cuff size adjusted to arm circumference). The average of the three measurements carried out at one-minute intervals was calculated.

Enzymatic methods were used to measure glycemia and triglycerides, as well as precipitation methods for HDL cholesterol after fasting for at least twelve hours. The measuring unit used was mg/dL. LDL was obtained indirectly using the Friedewald formula. Blood samples were obtained by peripheral venipuncture and sent to reference laboratories, where they were processed between 48 and 72 h.

To calculate Insulin Resistance, three risk scales were used:-Triglycerides/HDL. The cut-off point for insulin resistance was set at 2.4. [37]-Triglyceride glucose index (TyG index) [38] was calculated from the following formula:
TyG index = Ln(triglycerides × glycemia/2)

The cut-off point for the TyG index has been established at 8.8 in men and 8.7 in women [39].

Metabolic Score for Insulin Resistance (METS-IR) [40] was obtained using the formula:METS-IR = Ln(2 × glycemia + triglycerides) × BMI/Ln HDL-c

MetS was determined using three models:(a)NCEP ATP III [41] considers metabolic syndrome when there are three or more of the following events: blood pressure greater than 130/85 mmHg; triglycerides greater than 150 mg/dL or specific treatment for this lipid disorder; Low HDL and blood glucose less than 100 mg/dl or specific treatment for this glycemic disorder.(b)The International Diabetes Federation (IDF) [42] requires, in addition to two of the factors mentioned above for ATP III (triglycerides, HDL, blood pressure, and blood glucose), a waist circumference greater than 80 cm in women and 94 cm in men.(c)The JIS model [43] establishes criteria that are similar to those of NCEP ATPIII but with abdominal waist cut-off points similar to the IDF.

As obesity scales based on hip circumference, the following were calculated:-Abdominal volume index (AVI) [44] determined using the formula
AVI = [2 cm (waist)^2^ + 0.7 cm (waist-hip)^2^]/1000

-Body adiposity index (BAI) [45] which was obtained by applying the formula

BAI = (100 × hip circumference)/(height × √height) − 18

A person was considered a smoker if they had consumed at least one cigarette in the previous month or if they had stopped smoking less than a year before.

To evaluate dietary and physical activity habits, the Mediterranean diet adherence questionnaire and the International Physical Activity Questionnaire (IPAQ) were used, both self-administered.

The “Mediterranean Diet Adherence Questionnaire” is based on the PREDIMED study (MeDiet-Predimed), which consists of 14 questions scored with 0 or 1 point each and is used to evaluate adherence to the Mediterranean diet. Scores less than nine indicate poor adherence to the diet, while scores of nine or more indicate excellent adherence [46].

The International Physical Activity Questionnaire (IPAQ) [47] has demonstrated its usefulness in evaluating the physical activity that a person has carried out in the last seven days. It is a self-administered questionnaire of seven questions, which allows easy assessment of the frequency, duration, and intensity of physical activity carried out by classifying it into three levels: vigorous, moderate, and low physical activity.

Three categories of social classes were created based on the profession and the proposal of the group of social determinants of the Spanish Society of Epidemiology [48], which establishes three categories: Class I includes directors and managers, athletes and artists, university professionals, and qualified self-employed workers. Class II includes intermediate occupations and unskilled self-employed workers. Class III includes unskilled workers.

### Statistical Analysis

The mean and standard deviation of quantitative variables were calculated using Student’s *t* test. The chi-square test was used for qualitative variables. The usefulness of the BAI and AVI scales for predicting insulin resistance and metabolic syndrome was calculated by applying ROC curves, obtaining the area under the curve (AUC), and the cut-off points with their sensitivity, specificity, and Youden index. The SPSS 28.0 program was used to perform the statistical analysis, and a statistical significance level of *p* < 0.05 was accepted.

## 3. Results

Men made up 60.2% of the sample. The average age of the participants was slightly over 39 years (39.2 years in women and 39.8 years in men), with the majority group being between 30 and 49 years of age. The majority belonged to the most disadvantaged social class (class III) and had only a primary education. The average values of blood pressure, basal blood glucose, cholesterol, and its fractions, and triglycerides were within normal limits, although values were, on the whole, higher in men. Women, on the other hand, had a greater adherence to the Mediterranean diet and did more physical activity on a regular basis. A high percentage of the studied population were smokers (37.1% in men and 33.0% in women). The complete data can be consulted in Table 1.

Figure 2 and Table 2 present the results of the ROC curves to determine the predictive value of BMI (blue line), waist circumference (red line), BAI (purple line), and AVI (blue line) in relation to the three criteria evaluated to determine the MetS. It is observed that the areas under the curve are always higher for AVI. These values are practically the same for AVI compared to BMI and waist circumference. Additionally, they are very similar to those obtained with the BAI. Particularly noteworthy are the high AVI values to predict MetS by applying the IDF criteria, especially in women.

Table 3 details the AVI cut-off points to predict MetS according to the three models used, separated by sex. In the case of the IDF in women, a cut-off point was established at 13.70 with a sensitivity of 90.2, specificity of 90.0, and Youden index of 0.802, whereas in men, the cut-off point was established at 17.59 with a sensitivity of 83.6, specificity of 83.6, and Youden index of 0.672. Likewise, we found that the Youden index for AVI presents very similar results in the three models used, both in men and women, to those obtained with the BMI and waist circumference.

Figure 3 and Table 4 present the results of the ROC curves to determine the predictive value of BMI (blue line), waist circumference (red line), BAI (purple line), and AVI (blue line) with respect to the high values of the insulin resistance risk scales. In this case, the areas under the curve only present high values for both sexes when using the METS-IR formula for both AVI and BAI, with a very good result in women on both scales. This also coincides with the results obtained for BMI and waist circumference.

In Table 5, we see the cut-off points calculated for both scales. The AVI cut-off points for predicting high values of the insulin resistance risk scales in women were set at 13.12 with a sensitivity of 86.2, a specificity of 86.0, and a Youden index of 0.722. In men, the cut-off point was 17.59 with a sensitivity of 81.5, specificity of 81.1, and Youden index of 0.626.

The BAI cut-off points for predicting high values of the insulin resistance risk scales in women were set at 33.88 with a sensitivity of 87.9, a specificity of 86.9, and a Youden index of 0.748. In men, the cut-off point was 27.91 with a sensitivity of 79.9, specificity of 79.9, and Youden index of 0.598. These results are also similar to those obtained for BMI and waist circumference. Although in the case of BMI, they have a higher Youden index.

## 4. Discussion

In this large population sample, we studied 193,462 workers of both sexes between 18 and 69 years of age, all of them from different autonomous communities in Spain. Although the sample is somewhat larger in men, 60% compared to women, 40%, we believe that the large volume of the sample and having stratified it by sex offer valid results.

As this is a sample that comes from workers who attend company medical examinations, we could be committing healthy worker bias since sick workers do not regularly attend these [49]. However, in our study, we did not compare a pathology between healthy and sick workers, but rather the relationship between two formulas to measure the amount of body fat and insulin resistance and MetS. Neither was a pathology related to the job or influenced by a substance to which workers are exposed evaluated. We consider that workers belonging to multiple companies also constitute a representation of the general population. Although people with MetS or insulin resistance are susceptible to suffering from more pathologies [50], it is our belief that the large sample size and the particular research question (relationship between formulas) override this bias as the study was not a question of occupational epidemiology [51].

Another factor that can influence metabolic syndrome is aging since there are numerous publications that demonstrate a change in metabolism with aging and, consequently, an increase in metabolic syndrome with aging. However, the published literature refers to the population over 60 years of age [52]. In our work, only 5.5% of the population (3.0% of men and 2.5% of women) is 60 or older, so due to the enormous size of the sample, we have considered that it will not influence the results.

In our study, AVI has a good predictive value for MetS in both men and women with the three formulas used. However, the highest values occur when applying the IDF MetS criteria. When applying the ATPIII and JIS MetS criteria, the AUC for AVI is greater than 0.8 in both cases, and this result occurs in both men and women, which can be interpreted as a good test result for both sexes. In the case of the AUC corresponding to the IDF definition, the results are greater than 0.9, also for both sexes, which implies a very good test result, enabling us to affirm that there is a close relationship between AVI and the prediction of MetS according to IDF criteria. If we also focus on women, the AUC can be seen to be 0.961, which is very close to 0.97, which is accepted as an excellent result in the interpretation of results [53]. These values are practically the same as those obtained with BMI and waist circumference both in the AUC and in the Youden index; the usefulness of these indexes, as we have commented at the beginning of the article.

Regarding BAI, this also has a good predictive value in women since, for the three formulas used, the AUC is greater than 0.75, which enables the test results to be interpreted as good. However, with regard to the men in the study, none of the three formulas used to assess the MetS have an acceptable predictive value with BAI since its results are evaluated as a regular test. Therefore, they are not considered a recommended formula to predict MetS in men. These results are lower than those obtained with BMI and waist circumference in both the AUC and the Youden index, and therefore, it is not better than these indices to evaluate metabolic and cardiovascular risk in our population.

When establishing the AVI and BAI cut-off points to predict MetS, we found that in the three formulas used, the sensitivity and specificity are low in the case of BAI for both sexes, with a Youden index ranging between 0.283 and 0.395, thus leading us to rule out this formula as it could give many false results.

In the case of AVI, both the JIS and IDF formulas have good sensitivity and specificity in both sexes. Thus, in women, the cut-off for IDF was established at 13.70 with a Youden index of 0.802, and the cut-off for JIS at 12.72 with a Youden index of 0.628. In the case of men, the best value was also obtained with the IDF formula with a cut-off of 17.59 and a Youden index of 0.672. As in women, the JIS formula is also acceptable, in whom a cut-off of 17.32 was established with a Youden index of 0.604.

To evaluate insulin resistance, three risk scales were used: TyG index, TG/HDL, and METS-IR. We are aware that most studies recommend the use of HOMA-IR (Homeostasis Model Assessment of Insulin Resistance) [54]; however, since we did not have the fasting insulin values of the people in our sample, we opted for other validated methods, such as the aforementioned risk scales that are also recommended due to their low cost [55,56,57,58,59].

The predictive value of AVI and BAI on insulin resistance risk scales only offers an AUC greater than 0.75 with the METS-IR formula. These results are good in both men and women, although again, it is in women where it is closer to unity, bordering on excellence. However, when establishing the cut-off points, we found that although the AVI values were still good, the cut-off point for women was 13.12 with a Youden index of 0.722, while the cut-off point for men was 17.59 with a Youden of 0.626. In the case of BAI for men, the Youden index was 0.598, with a sensitivity of 79.9 and a specificity of 79.9, which, being around 80%, could be accepted as valid. When comparing these results with the AUC and the Youden index for BMI and waist circumference, we find that the results are practically identical, although somewhat higher in the case of AVI when applying the METS-IR formula.

One study carried out by Feng et al. in a Chinese population in 2007 evaluated the usefulness of different overweight and obesity scales to predict insulin resistance in 570 non-diabetic people [60]. In this case, the authors, unlike our study, directly measured insulin resistance using HOMA-IR. Statistical analyses showed that BAI had a lower AUC than AVI in predicting insulin resistance. After applying the Spearman test, insulin resistance was more strongly correlated with waist circumference and AVI in men and with BMI in women, while BAI showed a low correlation for both sexes.

In a cross-sectional study carried out on 981 Spanish adolescents (456 boys and 525 girls) [61,62], the validity of AVI to predict MetS calculated according to eight different criteria was determined, revealing great variability in the results depending on the criteria used. Analysis of the ROC curves showed that the AUC was greater than 0.80 in both boys and girls, but only when the diagnosis of MetS was established according to the IDF criteria. These data agree with those obtained in our study, although our population is adults.

Another study carried out in the Chinese population, published in 2017, in which the ability of different anthropometric indices to predict MetS in a population between 40 and 65 years of age was evaluated, concluded that BMI and AVI are better than other anthropometric indices to predict MetS in both sexes [63].

One study was carried out on 886 Colombian university students with an average age of slightly over 21 years. After assessing the ability of three BAI adiposity indices to predict MetS using the IDF criteria, the authors concluded that BAI was useful for predicting MetS and established the cut-off point at 30.34 (AUC = 0.720–0.863) for women and 27.83 (AUC = 0.726–0.873) for men [64]. These findings differ from our results since the AUC of BAI to predict MetS according to IDF criteria was less than 0.75, and in the case of women, although the AUC was 0.781, the Youden index was 0.395, which indicates low sensitivity and specificity.

In 2022, a cross-sectional study carried out on the Vietnamese population from June 2007 to July 2008 was published. Four thousand and seventy-one people between 25 and 74 years old were studied, using nine anthropometric indices to predict MetS. NCEP ATP III and IDF criteria, with modified central obesity cut-off values for Asians, were used to define MetS. All the parameters studied to define MetS were found to present higher AUC values when the IDF criteria were used compared to NCEP ATP III. Among them, AVI presented a high predictive value of MetS in both sexes when IDF criteria were used, with a cut-off point of 16.2 (AUC = 0.99) in men and 12.8 (AUC = 0.97) in women [65]. These results also agree with those of our study, although in our case, the AUC was somewhat higher for women than for men.

Chen et al. carried out a study in a population of 4767 Uyghurs (an ethnic group living in the northwestern regions of China) selected from the Cardiovascular Risk Survey carried out between October 2007 and March 2010, with the objective of identifying the best predictor of MetS in this population [66]. Several anthropometric parameters—BMI, WHtR, TGs/HDL-C, and BAI—were included to predict MetS according to IDF criteria, concluding that, in the Uyghur population, WHtR (AUC = 0.838) was the best predictor of MetS in men, whereas in women it turned out to be the TG/HDL-C ratio (AUC = 0.815). In the case of BAI, the AUC was 0.709 for men and 0.699 for women. These last results are similar to those found in our work.

The same authors as the previous paper carried out another study in the Chinese adult population, in this case, the Kazakh population (ethnicity from the mixture of primitive Turkic and Mongolian tribes) [67], in which 4094 people were included, assessing the predictive power of different anthropometric scales on MetS applying the IDF criteria. In this population, they found that the best predictor of MetS for both sexes was WHtR, with an AUC of 0.821 in men and 0.835 in women. In this case, the AUC for the TG/HDL-C index was 0.792 in men and 0.778 in women. Similar to the previous study, and in agreement again with our study, the AUC for BAI was 0.666 for men and 0.751 for women, which confirms that there is no good association between BAI and its predictive value for MetS.

### Strengths and Limitations

As a strength of the study, we would like to highlight the large sample size, which gives enormous power to the results obtained. As the sample comes from different companies with the largest number of workers in Spain, they constitute a representative sample of the population.

As a main limitation, it is worth noting that insulin resistance was not determined using an objective method but rather using risk scales. Another limitation of our study is not knowing the people in our population with previous treatments for metabolic disorders or macrovascular diseases.

## 5. Conclusions

Only AVI demonstrated its value as a predictor of metabolic syndrome and exclusively applied IDF criteria. This predictive value is higher in women.

AVI and BAI demonstrated their value as predictors of high values of insulin resistance risk scales only in the case of METS-IR. This predictive value is also higher in women.

## Figures and Tables

**Figure 1 diagnostics-13-03356-f001:**
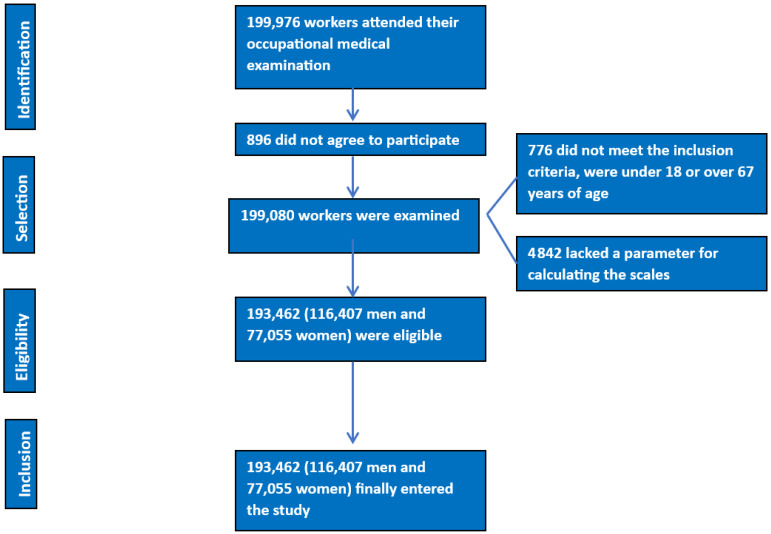
Flow chart of participants.

**Figure 2 diagnostics-13-03356-f002:**
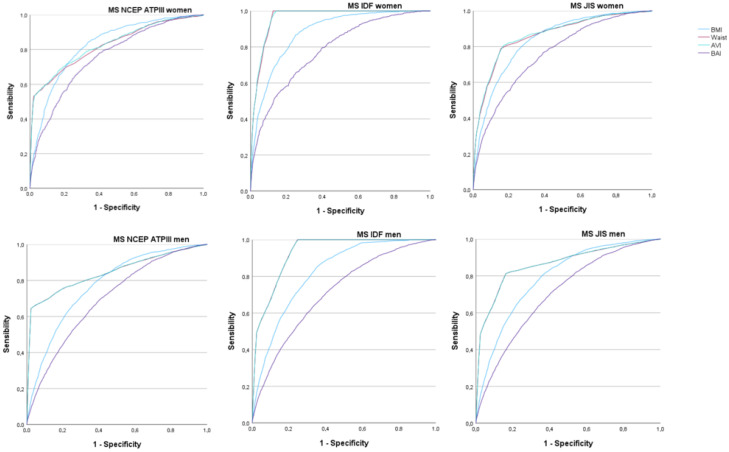
ROC curves to assess the usefulness of BAI and AVI for predicting metabolic syndrome by sex.

**Figure 3 diagnostics-13-03356-f003:**
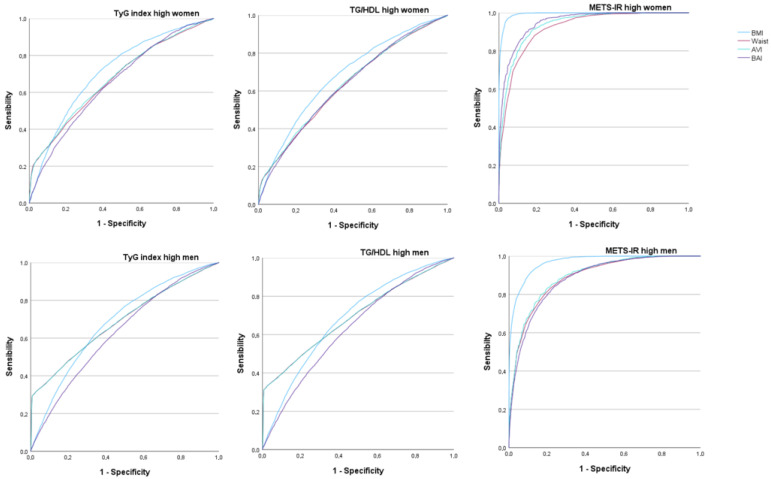
ROC curves to assess the usefulness of BAI and AVI for predicting high-risk values of insulin resistance scales by sex.

**Table 1 diagnostics-13-03356-t001:** Characteristics of the population.

	Men *n* = 116,407	Women *n* = 77,055	
	Mean (SD)	Mean (SD)	*p*-Value
Age (years)	39.8 (10.3)	39.2 (10.2)	<0.001
Height (cm)	173.9 (7.0)	161.2 (6.6)	<0.001
Weight (kg)	81.1 (13.9)	65.3 (13.2)	<0.001
Waist circumference (cm)	87.7 (9.1)	73.9 (7.9)	<0.001
Hip circumference (cm)	100.0 (8.4)	97.2 (8.9)	<0.001
Systolic blood pressure (mmHg)	124.4 (15.1)	114.4 (14.8)	<0.001
Diastolic blood pressure (mmHg)	75.4 (10.6)	69.7 (10.3)	<0.001
Total cholesterol (mg/dL)	195.9 (38.9)	193.6 (36.4)	<0.001
HDL-c (mg/dL)	51.0 (7.0)	53.7 (7.6)	<0.001
LDL-c (mg/dL)	120.5 (37.6)	122.3 (37.0)	<0.001
Triglycerides (mg/dL)	123.8 (88.0)	88.1 (46.2)	<0.001
Glycemia (mg/dL)	88.1 (12.9)	84.1 (11.5)	<0.001
	**%**	**%**	***p*-Value**
20–29 years	17.9	19.5	<0.001
30–39 years	33.1	33.3	
40–49 years	29.7	29.4	
50–59 years	16.3	15.3	
60–69 years	3.0	2.5	
Primary school	61.2	51.8	<0.001
Secondary school	34.0	40.7	
University	4.8	7.5	
Social class I	5.3	7.2	<0.001
Social class II	17.4	33.2	
Social class III	77.3	59.8	
No physical activity	54.5	47.8	<0.001
Yes physical activity	45.5	52.2	
Non-healthy food	59.0	48.6	<0.001
Healthy food	41.0	51.4	
Non-smokers	62.9	67.0	<0.001
Smokers	37.1	33.0	

HDL-c High-density lipoprotein cholesterol. LDL-c Low-density lipoprotein cholesterol.

**Table 2 diagnostics-13-03356-t002:** Areas under the BAI and AVI curve for predicting metabolic syndrome with the different criteria by sex.

	AVI Women	BAI Women	BMI Women	Waist Women
	AUC (95% CI)	AUC (95% CI)	AUC (95% CI)	AUC (95% CI)
MS ATPIII	0.829 (0.820–0.838)	0.757 (0.745–0.766)	0.825 (0.818–0.833)	0.824 (0.815–0.833)
MS IDF	0.961 (0.959–0.963)	0.781 (0.773–0.789)	0.880 (0.875–0.885)	0.960 (0.959–0.962)
MS JIS	0.866 (0.860–0.872)	0.759 (0.752–0.767)	0.837 (0.831–0.843)	0.862 (0.856–0.868)
	**AVI Men**	**BAI Men**	**BMI Men**	**Waist Men**
MS ATPIII	0.844 (0.839–0.849)	0.697 (0.692–0.702)	0.775 (0.771–0.780)	0.843 (0.838–0.848)
MS IDF	0.929 (0.927–0.930)	0.711 (0.707–0.716)	0.822 (0.819–0.825)	0.929 (0.927–0.931)
MS JIS	0.864 (0.860–0.868)	0.705 (0.701–0.710)	0.792 (0.788–0.795)	0.864 (0.860–0.867)

**Table 3 diagnostics-13-03356-t003:** Cut-off points with their sensitivity, specificity, and Youden index of BAI and AVI for predicting metabolic syndrome with the different criteria by sex.

	AVI Women	BAI Women	BMI Women	Waist Women
	Cutoff-Sens-Specif-Youden	Cutoff-Sens-Specif-Youden	Cutoff-Sens-Specif-Youden	Cutoff-Sens-Specif-Youden
MS ATPIII	12.34–73.9–73.8–0.477	31.62–69.5–69.3–0.388	27.25–75.8–74.7–0.505	77.0–74.1–71.5–0.457
MS IDF	13.70–90.2–90.0–0.802	31.62–69.8–69.7–0.395	27.95–79.2–79.1–0.583	82.0–90.0–89.9–0.799
MS JIS	12.72–81.6–81.2–0.628	31.46–68.8–68.7–0.375	27.25–76.6–75.9–0.529	78.50–81.0–81.0–0.620
	**AVI Men**	**BAI Men**	**BMI Women**	**Waist Women**
MS ATPIII	16.07–76.7–76.7–0.534	26.42–64.2–64.1–0.283	27.95–71.1–70.8–0.419	91.0–78.8–72.8–0.516
MS IDF	17.59–83.6–83.6–0.672	26.42–65.2–65.1–0.303	28.0–75.1–72.5–0.476	93.0–85.6–82.7–0.683
MS JIS	17.32–82.0–82.0–0.640	26.36–64.9–64.8–0.287	27.95–72.1–71.4–0.435	93.0–82.3–81.8–0.641

**Table 4 diagnostics-13-03356-t004:** Areas under the BAI and AVI curve for predicting high-risk values of insulin resistance scales by sex.

	AVI Women	BAI Women	BMI Women	Waist Women
	AUC (95% CI)	AUC (95% CI)	AUC (95% CI)	AUC (95% CI)
TyG index	0.678 (0.671–0.684)	0.655 (0.649–0.661)	0.713 (0.707–0.718)	0.673 (0.66–0.679)
TG/HDL	0.637 (0.632–0.642)	0.632 (0.627–0.637)	0.680 (0.676–0.685)	0.632 (0.627–0.637)
METS-IR	0.936 (0.933–0.939)	0.951 (0.948–0.953)	0.993 (0.993–0.994)	0.919 (0.916–0.923)
	**AVI Men**	**BAI Men**	**BMI Men**	**Waist Men**
TyG index	0.689 (0.686–0.693)	0.629 (0.625–0.633)	0.683 (0.679–0.686)	0.689 (0.685–0.693)
TG/HDL	0.694 (0.691–0.698)	0.633 (0.629–0.636)	0.682 (0.679–0.686)	0.694 (0.690–0.698)
METS-IR	0.894 (0.891–0.897)	0.882 (0.878–0.885)	0.968 (0.967–0.970)	0.889 (0.886–0.892)

**Table 5 diagnostics-13-03356-t005:** Cut-off points with their sensitivity, specificity, and Youden index of BAI and AVI for predicting high-risk values of insulin resistance scales by sex.

	AVI Women	BAI Women	BMI Women	Waist Women
	Cutoff-Sens-Specif-Youden	Cutoff-Sens-Specif-Youden	Cutoff-Sens-Specif-Youden	Cutoff-Sens-Specif-Youden
TyG index	11.64–61.8–61.6–0.234	30.42–61.1–61.0–0.221	25.65–66.6–66.3–0.329	75.0–61.8–61.1–0.229
TG/HDL	11.46–59.5–59.3–0.188	30.07–59.5–59.3–0.188	25.05–63.8–63.5–0.273	75.0–63.0–61.8–0.248
METS-IR	13.12–86.2–86.0–0.722	33.88–87.9–86.9–0.748	32.15–96.1–95.9–0.920	79.0–85.8–82.9–0.687
	**AVI Men**	**BAI Men**	**BMI Women**	**Waist Women**
TyG index	11.60–62.5–62.3–0.248	25.72–59.3–59.0–0.183	26.85–65.2–62.7–0.279	88.0–62.4–62.2–0.244
TG/HDL	15.64–62.5–62.4–0.249	25.75–59.7–59.2–0.189	26.85–65.5–62.2–0.277	88.0–62.8–61.8–0.246
METS-IR	17.59–81.5–81.1–0.626	27.91–79.9–79.9–0.598	30.45–89.9–89.9–0.798	93.0–81.6–79.9–0.615

## Data Availability

The study data are stored in a database that complies with all security measures at ADEMA-Escuela Universitaria. The Data Protection Delegate is Ángel Arturo López González.

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
