# Peer review of "Relationship between Abdominal Volume Index and Body Adiposity Index and Scales of Insulin Resistance and Metabolic Syndrome"

_diagnostics, 2023, doi:10.3390/diagnostics13213356_

Round 1

Reviewer 1 Report

Dear Authors I have read the manuscript and in my opinion it is very interesting work. And I would recommend the article to be published

However, I have some comments to the methods and some references.

1.MEthods - please describe any indexes/mathematic models/formulas as mathematic calculation not prescribed by world program - thus the formulas which for example must be squered wil be understabled.

2. The cited articles no 41, 42 are information from review articles, while in the given methods, the criteria for diagnosing metabolic disorders by ATPIII or IDF should be directly from the committies.

3 The results - while we age the metabolism change see doi: 10.2147/CIA.S306982 so as your population were from 18-67 it may influence the results.

What is more I recomedn to perform the multiple regression analysis to exclude the potentials co-factors as age sex ect.

Minor editing of English language required

Author Response

Reviewer 1

Dear reviewer,

First of all, thank you for your work and all your recommendations.

To facilitate your review, we have written the modifications in red in the article.

Comments and Suggestions for Authors

Dear Authors I have read the manuscript and in my opinion it is very interesting work. And I would recommend the article to be published

However, I have some comments to the methods and some references.

1.MEthods - please describe any indexes/mathematic models/formulas as mathematic calculation not prescribed by world program - thus the formulas which for example must be squered wil be understabled.

Dear reviewer. We are very sorry that we are not able to understand what you are asking of us. We would ask you to express it in another way or give us an example. Thank you so much.

  1. The cited articles no 41, 42 are information from review articles, while in the given methods, the criteria for diagnosing metabolic disorders by ATPIII or IDF should be directly from the committies.

Thank you very much for your recommendation. We have proceeded to modify the citations as you have indicated.

  1. Expert Panel on Detection, Evaluation, and Treatment of High Blood Cholesterol in Adults. Executive Summary of The Third Report of The National Cholesterol Education Program (NCEP) Expert Panel on Detection, Evaluation, And Treatment of High Blood Cholesterol In Adults (Adult Treatment Panel III). JAMA. 2001 May 16;285(19):2486-97. doi: 10.1001/jama.285.19.2486.
  2. The IDF consensus worldwide de¬ nition of the metabolic syndrome. Available at: https://idf.org/media/uploads/2023/05/attachments-30.pdf#:~:text=The%20new%20IDF%20defi%20nition%20addresses%20both%20clinical,studies%20and%20other%20research%20into%20the%20metabolic%20syndrome.

3 The results - while we age the metabolism change see doi: 10.2147/CIA.S306982 so as your population were from 18-67 it may influence the results.

Dear reviewer, you are right. We have reviewed the article that was suggested to us and we have seen that all the works refer to the population over 60 years of age. In our work, only 5.5% of the population (3.0% of men and 2.5% of women) is 60 or older, so due to the enormous size of the sample we have considered that it will not influence in the results.

We have made the justification and have cited the article that has referenced us that we consider to be of vital importance in the discussion of our work.

Thank you so much.

What is more I recomedn to perform the multiple regression analysis to exclude the potentials co-factors as age sex ect.

Many thanks for your recommendation. In our study we have not performed multiple regression, because as we have mentioned previously we consider that due to the high sample size and the low representation of the age over 60 years it has no influence on the results, and is not the objective of the study. study to know the effect of age and sex. The article is part of a doctoral thesis in which the usefulness of the Abdominal Volume Index and Body Adiposity Index in predicting cardiometabolic alterations is assessed.

However, if you consider it essential to carry out such analysis for the publication of the article, let us know and we will do it.

Thank you very much for your suggestions. We have proceeded to answer all of them and we trust that they will adequately respond to your questions.

Reviewer 2 Report

This study examined the relationship between MetS, insulin resistance index, and abdominal obesity index in a large general Spanish sample.

While the results are valid, there are several concerns regarding the introduction and methodology.

- The Introduction is overly descriptive of the physiological effects of insulin.

- The rationale for the selection of TG/HDL-C, TyG index, and METS-IR as risk indicators for insulin resistance is not provided.

- The rationale for the selection of only AVI and BAI as abdominal obesity indices is not provided. Many abdominal obesity indices have been reported, for example, WC/BMI ratio, WHtR, Body roundness index, Conicity index, and ABSI.

- A history of treatment for metabolic disorders or macrovascular disease was not noted. These should have a significant impact on the results.

- Drinking alcohol increases TG independently of insulin resistance, and information on the proportion of drinkers seems essential if the TyG index is to be evaluated.

- How was the comparison of predictive ability in ROC analysis determined: AUC, NRI or IDI?

- The authors emphasized at the beginning the usefulness of the abdominal obesity index compared to BMI and WC. Therefore, when examining the association with MetS and insulin resistance, not only AVI and BAI but also BMI and WC must be analyzed simultaneously.

Author Response

Dear reviewer,

First of all, thank you for your work and all your recommendations.

To facilitate your review, we have written the modifications in red in the article.

Reviewer 2

Dear reviewer,

First of all, thank you for your work and all your recommendations.

To facilitate your review, we have written the modifications in red in the article.

This study examined the relationship between MetS, insulin resistance index, and abdominal obesity index in a large general Spanish sample.

While the results are valid, there are several concerns regarding the introduction and methodology.

- The Introduction is overly descriptive of the physiological effects of insulin.

Following your suggestions, we have shortened the description of the physiological effects of insulin.

- The rationale for the selection of TG/HDL-C, TyG index, and METS-IR as risk indicators for insulin resistance is not provided.

These three indices are among the most used in the literature to assess the risk of IR. Furthermore, they are the only ones that can be carried out on such a large sample size due to their low cost. HOMA has not been used since they require insulin values that were not available and also its cost would be unaffordable in such a large sample. This justification is found in the eighth paragraph of the discussion.

- The rationale for the selection of only AVI and BAI as abdominal obesity indices is not provided. Many abdominal obesity indices have been reported, for example, WC/BMI ratio, WHtR, Body roundness index, Conicity index, and ABSI.

These two obesity indices have been selected because the article is part of a doctoral thesis, in which the usefulness of these two indices in predicting cardiometabolic alterations is assessed.

- A history of treatment for metabolic disorders or macrovascular disease was not noted. These should have a significant impact on the results.

You are right. However, in our database we do not have treatments for these pathologies, so we cannot record them. If it seems good to you, we include it in the weaknesses of the study.

- Drinking alcohol increases TG independently of insulin resistance, and information on the proportion of drinkers seems essential if the TyG index is to be evaluated.

To avoid the influence of alcohol on TG. People who were habitual drinkers in our database were excluded. Thank you very much for your observation

- How was the comparison of predictive ability in ROC analysis determined: AUC, NRI or IDI?

 To assess the predictive capacity in the ROC analysis, AUC was assessed.

- The authors emphasized at the beginning the usefulness of the abdominal obesity index compared to BMI and WC. Therefore, when examining the association with MetS and insulin resistance, not only AVI and BAI but also BMI and WC must be analyzed simultaneously.

Thank you very much for your recommendation. We have proceeded to include both BMI and WC in the ROC curves. Just as we have created the new AUC, cutoff and Youden index tables. Both in predicting metabolic syndrome and in predicting high-risk values of insulin resistance.

Thank you very much for your suggestions. We have proceeded to answer all of them and we trust that they will adequately respond to your questions.

Round 2

Reviewer 2 Report

The authors have addressed the previous review comments to the extent possible.

The reviewer generally has no objection to the revised manuscript. On the other hand, the following items require minor revisions before the accept.

Ø  (Line302-305) When comparing these results with the AUC and the Youden index for BMI and Waist circumference, we find that the results are practically identical, although somewhat higher in the case of AVI when applying the METS-IR formula.

Really? According to Table 4, doesn't BMI have the highest predictive ability for MET-IR?

AVI/BAI did not necessarily have better predictive ability for insulin resistance index or MetS diagnosis compared to BMI/WC. What outcome should be set in the future to confirm the superiority of AVI/VAI? This discussion should be added.